# OpenReview forum: "AutoBencher: Towards Declarative Benchmark Construction"
_ICLR.cc/2025/Conference — ICLR 2025 Poster_

### Official Review · Reviewer_vFMj · 2024-10-31

**Soundness:** 3
**Presentation:** 4
**Contribution:** 3
**Rating:** 6
**Confidence:** 3

**Summary:**

This paper introduces AutoBencher, a declarative framework that automatically generates benchmarks to evaluate language model performance by defining specific desiderata, such as difficulty, novelty, and safety. The proposed approach, AutoBencher, leverages language models to iteratively construct benchmarks that reveal weaknesses and safety vulnerabilities in LLMs.

**Strengths:**

1-The paper is well-written and very easy to understand.

2- The paper provides an in-depth empirical evaluation, uncovering vulnerabilities and assessing model performance. It also includes results from human evaluations to further validate the quality and relevance of the generated benchmark.

3- AutoBencher automatically constructs datasets, making it highly scalable and reducing reliance on costly human labor. This is useful for evaluating LLMs across various domains.

**Weaknesses:**

1- The details of the qualitative analysis are missing.

2- AutoBencher uses GPT-4 as the evaluator, which may introduce potential bias in datasets that could favor LLMs from the same family (e.g., OpenAI's LLMs).

**Questions:**

1- How does AutoBencher differ from previous work on automatic benchmarking, such as that in [2]?

2- Why is a comparison with the baseline included?





[1] Taskbench: Benchmarking large language models for task automation.

---

> ### Author Response · Authors · 2024-11-22
>
> We thank reviewer vFMj for their time and review. Reviewer vFMj primarily asks for ablation studies justifying the impact of different evaluator LM, and clarifications on the qualitative analysis. We have added additional discussions in the paper to address these, summarized below. Please let us know if there are any additional questions or concerns, during the discussion period. Thank you!
>
> **“AutoBencher uses GPT-4 as the evaluator, which may introduce potential bias in datasets that could favor LLMs from the same family (e.g., OpenAI's LLMs).”**
>
> For all experiments in the paper, we use GPT-4-turbo as the evaluator LM. We notice that GPT-4-turbo generated questions induce the following model ranking:  claude-3-opus-20240229 > gpt-4-turbo-2024-04-09 > claude-3-sonnet-20240229 > gpt-3.5-turbo. Since claude-3 is ranked the highest, it suggests that GPT-4 is not exactly biasing towards models in its family. To further justify this point, we set the evaluator LM as Claude-3.5-sonnet.
>
> |                                | Novelty       | Separability       | Difficulty        |
> |--------------------------------------|---------------|--------------------|-------------------|
> | **MMLU (HumanBench)**                | 0.13          | 0.011             | 0.056            |
> | **AutoBencher (EvalLM=Claude-3.5)**  | 0.432         | 0.034             | 0.591            |
> | **AutoBencher (EvalLM=gpt-4-turbo)** | 0.43          | 0.026             | 0.321            |
>
> We find that the dataset discovered by Claude reveals the same relative ranking of the GPT and Claude families, suggesting that the evaluator LM is not significantly biased towards models in its family. Moreover, we report the novelty, separability, and difficulty score of AutoBencher-claude, it’s similar to AutoBencher-GPT4 in novelty and separability, slightly better in difficulty, and preserves the trend compared with HumanBench.
>
> For the safety evaluation setting, we set the evaluator LM as LLaMA 3.1-405B, and we report the ASR (attack success rate) below.  We find that AutoBencher (LLaMA-3.1-405B) attains a similar ASR as AutoBencher (gpt-4-turbo).
>
>
> |                              | ASR |
> |------------------------------------|--------|
> | Baseline (Harmbench)               | 0.284  |
> | AutoBencher (LLaMA 3.1-405B)       | 0.389  |
> | AutoBencher (gpt-4-turbo)      | 0.387  |
>
>
> **“How does AutoBencher differ from previous work on automatic benchmarking, such as TaskBench?”**
>
> TaskBench proposes a method for generating tool-use benchmarks. Specifically, they (i) construct a full tool graph, which captures all the dependencies between various tools; (ii) they sample a subgraph and use back-instruct to generate instructions that correspond to the subgraph. (iii) they verify the generated instructions and filter low-quality ones.
>
> AutoBencher offers a broader framing than TaskBench and they are complementary. TaskBench step (ii, iii) could be plugged into AutoBencehr as a module of automatic dataset generation for tool-use domain. In AutoBencher terminology, the back-instruct from the subgraph is privileged information that makes the question generation easier for the evaluator LM. This module would be parallel to our automatic dataset generation for knowledge intensive / math QA domains.
>
> We can use the optimization loop in AutoBencher to augment TaskBench by (i) brainstorming more tools and proposing a diverse set of tool graphs.  For example, AutoBencher can construct tool graphs based on task descriptions, and it could form new graphs that combine “Huggingface” tools and “Multimedia” tools. (ii) search for systematic failures in tool-use. For example, AutoBencher could discover that the model fails for the chain length exceeding 5; or maybe the model fails when it involves audio-generation followed by retrieval.

---

> > ### Author Response · Authors · 2024-11-22
> >
> > **“The details of the qualitative analysis are missing.”**
> >
> > We thank the reviewer for pointing this out, and we have added more details in the revised version of the paper. In the qualitative analysis section, the main result is in table 2 and table 12. We list some dataset descriptions discovered by AutoBencher, and we randomly draw a few (question, answer) pairs from each dataset. For the capability evaluations: science, economics and math, we list the question and the gold answer generated by the evaluator LM. For the safety evaluation, we list the harmful question, the unsafe answer generated by the jailbroken candidate LM because the gold answer should always be to abstain.
> >
> > In the main text, we discuss the ranking of candidate models on three capability datasets and two safety datasets generated by AutoBencher. For example, Gemini Pro attains a general ranking of 6 on MMLU, but it drops to 16 on Fordism, explaining why it is novel performance patterns not captured by existing datasets. Another example we analyzed is safety datasets that successfully jailbreak most of candidate LMs, we pointed out the systematic pattern in that dataset, and hypothesized the reason for successful jailbreak.
> >
> >
> >
> >
> > **“Why is a comparison with the baseline included?”**
> >
> > We compare against the baseline of human-written or crowdsourced benchmarks (e.g., MMLU, HarmBench), and report the desiderata metrics on these benchmarks. The reason we include this comparison is to show that AutoBencher-capability datasets are more difficult, novel and separable; and AutoBencher-safety datasets reveal more safety vulnerabilities than the human-generated counterparts. These comparisons demonstrate the usefulness of AutoBencher, beyond simply reducing costly human labor.

---

### Official Review · Reviewer_oakz · 2024-11-03

**Soundness:** 4
**Presentation:** 4
**Contribution:** 4
**Rating:** 8
**Confidence:** 3

**Summary:**

The authors propose to use an LM-guided adaptive search to construct benchmarks that optimise (1) difficulty (benchmark headroom), separability (existing model variance), and novelty (rank correlation with other benchmarks) and (2) attack success rate (when trying to extract harmful information from models). They demonstrate the need for this by showing that existing benchmarks exhibit low values in novelty (especially) even for benchmarks such as MMLU where there is sufficient headroom. Further, they use privileged information to enable a weaker LM to assess a stronger LM, use translation tools to enable multi-lingual data and employ source code/python to evaluate numerical and symbolic expressions. When optimising, the authors employ an adaptive search strategy that uses the history of explored subjects to guide the new candidates restricted to a salience set. To assess the quality of the generated data, the authors employ Mechanical Turk and find sufficiently low error rates with high salience on the questions. The value of adaptive search is demonstrated via an ablation on the use of the history vector. Both regimes demonstrate improvement over HumanEval on the desired metrics.

**Strengths:**

+ On novelty, the approach generates human interpretable topics where model ranks exhibit surprising results.
+ On safety, the qualitative examples align with the experience of this reviewer when trying to manually jail-break LLMs: pose the question as a hypothetical or philosophical debate; this vector being auto-discovered is encouraging.
+ Well-executed research methodology with manual validation of the discovered benchmarks

**Weaknesses:**

- The approach to the math and science categories suggests an open vocabulary problem that is not clearly tackled. For other categories, this is tackled via Wikipedia and a popularity metric.
- The translation induces an issue for low-resource language: such languages are both less likely to be tackled by LLMs and by translation tools, creating a catch-22. (I feel it would be sufficient to acknowledge the issue as a limitation since tackling the issue requires significant manual effort, future work can consider specialised models for specific language pairs.)

**Questions:**

1. To the knowledge of this reviewer, it should be possible to have a hyper-/hyponymy graph of Wikipedia topics. Did you consider using such a graph, or a similar topic relationship graph to further guide the adaptive search?
2. The optimisation problem for case (1) is multi-objective, is there a reason why multi-objective optimisation was not employed and instead the loss is linearised by adding hyperparameters? (evolutionary/population-based methods?)
3. Related to W1, how is the math/science salience set constructed? Unless I missed it in the paper, the inclusion-exclusion criteria were not clear or indeed the exact methodology. Did you perform, for example, open card sorting or a similar methodology?

**Details Of Ethics Concerns:**

There are ethical issues with the topic under study as it enables attack vectors on LLMs; however, the authors take due diligence to properly discuss their study process and acknowledge the broader impact.

---

> ### Author Response · Authors · 2024-11-21
>
> We thank reviewer oakz for their time and review. Reviewer oakz primarily asks for clarifications on the salience criteria. We have added additional discussion in the paper to address these, summarized below. Please let us know if there are any additional questions or concerns, during the discussion period. Thank you!
>
>
> **“how is the math/science salience set constructed? The approach to the math and science categories suggests an open vocabulary problem that is not clearly tackled.”**
>
> For science, we still use the number of page views of the corresponding Wikipedia article. For math, we evaluate the categories by including humans in the loop to judge and prune the non-salient categories. Specifically, we didn’t do open card sorting, we manually classify each description into two categories {salient, non-salient}. Empirically, we observe that the rejection rate for math categories is close to 0, because prompted GPT-4 for math categories yields highly salient math categories.
>
>
> **“Low-resource languages are both less likely to be tackled by LLMs and by translation tools”**
>
> We agree with the reviewer that low-resource languages are a problem that’s hard to address, and we have added this discussion in the limitation section.
>
>
> **"hyper-/hyponymy graph of Wikipedia topics. Did you consider using such a graph, or a similar topic relationship graph to further guide the adaptive search?"**
>
> Yes, we have considered a similar idea of using categories and subcategories in Wikipedia in a preliminary experiment.  Empirically, for each category (e.g., physical science), there is a large number of subcategories (on a scale of thousands) at different granular levels. We considered concatenating all these category names in the context of the language models to guide the selection of next topics, but this is computationally expensive. Further, this doesn’t help improve dataset desiderata, as the language model often ends up picking some non salient subcategories.
>
> **“why multi-objective optimisation was not employed and instead the loss is linearised by adding hyperparameters? (evolutionary/population-based methods?)”**
>
> Thanks for pointing out alternative methods of multi-objective optimization. In our use case, linearizing the multiple losses  is a good simplification, because we don’t need to construct the full pareto frontier, instead we are just looking for one dataset that maximizes the specified desiderata. The evolutionary algorithms could still be a viable alternative to iteratively optimize our linearized objective.

---

> > ### Comment · Reviewer_oakz · 2024-11-22
> >
> > Thank you for your clarifications, I feel most of my concerns are addressed; however, I have a small follow-up on the math category.
> >
> > The clarification that it is a binary classification rather than manual labelling removes the open vocabulary issue, since now any new category is classified as either salient or not. However, the manual process still raises the question of inter-rater agreement or whether there is a moderation protocol to sort out disagreements between raters. (To clarify, this is strictly about the math salience set, not the MT experiment. I _do not_ expect such measures on the MT data)

---

> > > ### Author Response · Authors · 2024-11-25
> > >
> > > Thanks for the follow-up question!
> > >
> > > For this paper, we didn't query multiple annotators. We agree that inter-annotator disagreement might exist for math. We can propose two ways of addressing this (1) collect 5 annotations and tie-break based on majority votes and (2) tie-break based on expert opinions. Especially for these domain-specific questions, expert opinions (e.g., PhD in the field) with rationales could provide a trustworthy perspective.

---

> > > > ### Comment · Reviewer_oakz · 2024-11-26
> > > >
> > > > Thank you for the clarification.
> > > >
> > > > Either (1) or (2) would address the issue, and I lean towards (2) given the nature of the data. A single annotator introduces an unknown bias even when the single annotator is an expert.

---

### Official Review · Reviewer_bP3L · 2024-11-08

**Soundness:** 3
**Presentation:** 3
**Contribution:** 3
**Rating:** 6
**Confidence:** 2

**Summary:**

The paper presents a method, Autobencher, to automatically construct benchmarks for evaluating language model (LM) capabilities and safety. Specifically, based on predefined criteria, the authors prompt an LM to propose dataset descriptions and generate questions, which a candidate LM then attempts to answer. The dataset is subsequently scored according to these criteria, enabling the selection of high-quality dataset descriptions. The authors aim to demonstrate that Autobencher generates dataset topics/descriptions that are both more novel and more challenging than existing benchmarks.

**Strengths:**

- The paper conducts experiments across a wide range of knowledge domains.
- The presentation is generally clear, with well-structured sections and a logical flow.
- Equations and figures are used effectively to enhance clarity.
- The automatic generation of benchmarks is a topic of strong interest to the community.
- The proposed method demonstrates significant performance gains over human-generated benchmarks, particularly in terms of novelty, difficulty, and separability metrics as defined in the paper.

**Weaknesses:**

- It is not entirely clear to me what methodological novelties the paper introduces in dataset construction, aside from its use of retrieval during generation. Optimizing for various criteria objectives is not a particularly significant contribution.
- The introduction and abstract could benefit from revisions to be more specific and descriptive.
- The method and criteria rely on accuracy scores computed across both existing models and preexisting datasets, which can be computationally intensive, especially if generating multiple datasets across a large set of models.
- Additionally, language models (LMs) can exhibit high variance across different runs. A robust scoring criterion should account for this issue by incorporating variance between different runs, both for generating questions and answers. Introducing a statistical method to handle this variance would strengthen the contribution.
- Ensuring that the dataset description is clear and unharmful does not guarantee that all generated questions are equally clear and unharmful. For instance, a dataset labeled as "Nuclear technology" might contain questions of varying levels of harmfulness.
- There are concerns regarding the practicality of the method. While the paper claims high automation, many of the criteria still seem to require potentially time-intensive manual crafting. Furthermore, LM generations can be compute-intensive; a discussion on the computational resources required for these generations would be beneficial.
- Additional ablation studies could help demonstrate the impact of various components of the method.

**Questions:**

- In Section 3.1 on separability, are you taking the mean over $acc(\cdot) \in V_c$? If so, this should be clarified. Additionally, I am not convinced that high deviation alone ensures that rankings between any two models, $LM_i$ and $LM_j$, with $i \ne j$, are robust to noise.
- Using vector notation could help make the equations clearer.
- In Section 3.1 on novelty, it seems that there is a coefficient for each model being evaluated. Could you elaborate on why there are coefficients for each model rather than for each dataset?
- How are the salience and harmfulness binary variables obtained in practice?
- I am unclear about the procedure for qualitative assessment in Section 6.3. What does "high quality" mean, and how many question-answer pairs were sampled? What do the "ranks" in this section refer to?
- What language model was used to generate Table 1? Adding this information to the caption would improve clarity.
- Are there any ablation experiments on the following: the language model used, the criteria for evaluating dataset descriptions, and the impact of including privileged information?
- Are the human benchmarks also grouped into distinctive categories and descriptions similar to those proposed by the language model?

---

> ### Author Response · Authors · 2024-11-21
>
> We thank reviewer bP3L for their time and review. Reviewer bP3L primarily asks for ablation studies of AutoBencher to understand the impact of (i) setting the temperature to 1, (ii) privileged information, (iii) different evaluator LMs; and clarifications on the practicality/compute of AutoBencher.  We have added several new experiments and additional discussion in the paper to address these, summarized below. Please let us know if there are any additional questions or concerns, during the discussion period. Thank you!
>
>
> **“language models (LMs) can exhibit high variance across different runs. A robust scoring criterion should account for this issue by incorporating variance between different runs, both for generating questions and answers. Introducing a statistical method to handle this variance would strengthen the contribution.”**
>
> In AutoBencher, there are two components that are subject to randomness, (1) dataset description proposal (2) (question, answer) generation. For all experiments in the paper, we set the decoding temperature for the evaluator LM to 0, which yields a deterministic response. We believe that the reviewer made a good point about building a better understanding of the variance involved in AutoBencher, therefore we added ablation studies to answer this question. We included this analysis in appendix G.
>
> First, we set temperature to 1 for the generation of (question, answer) pairs. This means that conditioned on the dataset description and privileged information, we could draw different QA pairs for the distribution. We report the Pearson correlation between the accuracy vectors below:
>
> Pearson Correlation Matrix:
> |         | seed1 | seed2 | seed3 |
> |---------|-------|-------|-------|
> | **seed1** |  1.00 |  0.96 |  0.98 |
> | **seed2** |  0.96 |  1.00 |  0.96 |
> | **seed3** |  0.98 |  0.96 |  1.00 |
>
>
>
> The Pearson correlation across different random seeds is close to 1, which means that the model accuracies are very similar across datasets generated with different random seeds. This suggests that our dataset generator is low variance and robust.
>
>
> Then, we extend this setting, and set temperature to 1 for proposing the dataset description. We find that randomness here leads to the discovery of different dataset descriptions. The new AutoBencher run generates the description:  “International Trade disputes on rare-earth elements”. Although this description is new, we attain similar levels of novelty and difficulty as our initial run, as seen below.
>
> | Datasets                    | Novelty       | Separability       | Difficulty         |
> |---------------------------|---------------|--------------------|--------------------|
> | **MMLU (HumanBench)**     | 0.13          | 0.011             | 0.056             |
> | **AutoBencher (temp=0)**  | 0.43          | 0.026             | 0.321             |
> | **AutoBencher (temp=1)**  | 0.383 ± 0.05  | 0.036 ± 0.02      | 0.301 ± 0.04      |
>
>
> These additional experiments support the argument that **AutoBencher is robust to variations in decoding temperature, as well as run-to-run variability when sampling**.

---

> > ### Author Response · Authors · 2024-11-22
> >
> > **“Additional ablation studies could help demonstrate the impact of various components of the method.”**
> >
> > We thank the reviewer for suggesting ablation studies, and we included three new ablations: (i) ablating the privileged information, (ii) ablation at temperature 1, (iii) ablation with different evaluator LM. We included these experiments in the revision in section G, H, I.
> >
> > (i) In AutoBencher, we leverage privileged information to create asymmetry between the evaluator LM and the candidate LMs, thereby generating higher quality questions that’s more difficult. In this ablation, we generate the questions without the privileged information. Specifically, we pick knowledge-intensive “economics” as the domain, and generate the questions without retrieving wikipedia articles.
> >
> > As shown in the table, the difficulty score is 0.0, meaning that the dataset (generated by GPT-4-turbo) is saturated by both claude-3-opus-20240229 and gpt-4-turbo-2024-04-09.  In fact, the median model performance on this dataset is 0.9523, which means that it’s hard to separate model accuracies.
> >
> > (ii) We perform ablation studies setting the temperature of the evaluator LM to 1.0. As shown in the table (capability), the general trend is very robust: our AutoBencher constructed dataset is more novel, separable and difficult than the human-written benchmarks. Also, the scores for each metric are similar to AutoBencher (temperature=0).
> >
> > As shown in the table (safety),  the high temperature experiment yields slightly lower ASR (0.356 v.s 0.387). Specifically, Autobencher (temperature=1.0) has difficulty identifying hard safety categories on the Claude family, resulting in a lower average ASR.
> >
> > (iii) For all experiments in the paper, we use GPT-4-turbo as the evaluator LM. We notice that GPT-4-turbo generated questions induce the following model ranking:  claude-3-opus-20240229 > gpt-4-turbo-2024-04-09 > claude-3-sonnet-20240229 > gpt-3.5-turbo. Since claude-3 is ranked the highest, it suggests that GPT-4 is not significantly biased towards models in the GPT-4 family. To further investigate, we set the evaluator LM as Claude-3.5-sonnet. We find that the discovered dataset reveals the same relative ranking of the GPT and Claude families, suggesting that the evaluator LM is not significantly biased towards models in its family. Moreover, we report the novelty, separability, and difficulty score of the AutoBencher-claude, it’s similar to AutoBencher-GPT4 in novelty and separability, slightly better in difficulty, and preserves the trend compared with HumanBench.
> >
> > For the safety setting, we experiment with evaluator LM as LLaMA 3.1-405B (see results below), and find that AutoBencher (LLaMA-3.1-405B) attains a similar ASR as AutoBencher (gpt-4-turbo). This ablation studies suggest that AutoBencher is quite robust to the choice of evaluator LM, and state-of-the-art LMs such as gpt-4, claude-3.5 and llama-405B can all serve as the evaluator LM.
> >
> > | capability (Econ)                             | Novelty       | Separability       | Difficulty        |
> > |--------------------------------------|---------------|--------------------|-------------------|
> > | **MMLU (HumanBench)**                | 0.13          | 0.011             | 0.056            |
> > | **AutoBencher (-privileged info)**   | 0.267         | 0.004             | 0.0              |
> > | **AutoBencher (EvalLM=Claude-3.5)**  | 0.432         | 0.034             | 0.591            |
> > | **AutoBencher (temperature=0)**      | 0.43          | 0.026             | 0.321            |
> > | **AutoBencher (temperature=1)**      | 0.383 ± 0.05  | 0.036 ± 0.02      | 0.301 ± 0.04     |
> >
> >
> > |     Safety                      | ASR |
> > |------------------------------------|--------|
> > | Baseline (Harmbench)               | 0.284  |
> > | AutoBencher (LLaMA 3.1-405B)       | 0.389  |
> > | AutoBencher (temperature ≈ 0)      | 0.387  |
> > | AutoBencher (temperature = 1)      | 0.356  |

---

> > > ### Author Response · Authors · 2024-11-22
> > >
> > > **“The method relies on accuracy computed across both existing models and preexisting datasets and LM generations [of questions] can both be compute-intensive. A discussion on the computational resources would be beneficial”**
> > >
> > > Overall, in the AutoBencher pipeline, there are two components that require compute:  (i) using evaluator LM to generate the datasets (ii) evaluating the candidate LMs on the generated datasets. We will discuss the compute cost for each component and we also added this discussion to Appendix F.
> > >
> > > As mentioned in Costs in section 5.2: “Each run of the AutoBencher agent uses around 750K tokens, which costs around $15. Among them, 43K tokens are used for proposing topics, 576K tokens are used for constructing datasets, and 147K for evaluating the candidate LM.” This dataset construction cost is more cost efficient when compared with expert-curated datasets, which often cost thousands of dollars.
> > >
> > > Regarding the cost to evaluate all the candidate LMs on both existing datasets and our new datasets, the computational cost is also moderate. First, accuracies on pre-existing datasets are publicly available online (e.g., HELM, OpenLLM Leaderboard), so they can be reused at zero cost. Second, there are two places where we evaluate the candidate models on our AutoBencher generated datasets: dataset selection and final evaluation of the selected dataset.
> > >
> > > For dataset selection, we generate a small dataset (|D| = 50) for each dataset description to reduce the cost (line 333 in the paper, line 6 and 12 in Algorithm 1), and there are roughly 20 descriptions for each AutoBencher run. The final evaluation on the selected dataset roughly involves |D| ~= 500 queries and 17 models. We use vLLM for model inference, and API calls for LLM-as-judge. We observe that LLM-as-judge is the actual compute time bottleneck, but this part can be parallelized significantly across models and across queries. As a result, our implementation is very time-efficient, it takes around 1h on 1 A100 GPU, and 30 dollars on API calls for dataset selection and 30 min on 1 A100 GPU, and 15 dollars on API calls for the final evaluation of 17 models. This is not computationally expensive given that we evaluated 17 models.
> > >
> > > **“what methodological novelties the paper introduces in dataset construction, aside from its use of retrieval during generation. Optimizing for various criteria objectives is not a particularly significant contribution”**
> > >
> > > Optimization, despite being commonly used in learning, is seldom used for dataset construction. The most related work to ours is automated red teaming, where the goal is to optimize for adversarial questions that maximize harmfulness, but AutoBencher generalizes this to be a broader set of desiderata such as “novelty, difficulty, separability, ASR, salience”, and extends the application to both capability and safety.
> > >
> > > Our work also contains empirical observations of high correlations between benchmarks (novelty) and that optimization-based benchmark construction can find benchmarks that behave orthogonally to the existing set of evaluations.
> > >
> > > Finally, on a methodological level, we study and identify effective search strategies such as incremental revision and the use of asymmetric test-taker and test-creator models that enable us to find much harder benchmarks than a naive rejection-sampling only approach would (see our ablations).
> > >
> > > **“ensuring that the dataset description is clear and unharmful does not guarantee that all generated questions are equally clear and unharmful. For instance, a dataset labeled as "Nuclear technology" might contain questions of varying levels of harmfulness.”**
> > >
> > > Recall in line 455 (section 6.4, paragraph 3), “we perform a human evaluation to validate the harmfulness of the generated questions. We found that 98.4% of the questions are labeled as harmful, and language models should abstain”. Specifically, we not only ensure the dataset descriptions are harmful but also verify that each question is indeed harmful under our safety criterion.
> > >
> > > **“While the paper claims high automation, many of the criteria still seem to require potentially time-intensive manual crafting.”**
> > >
> > > AutoBencher requires some manual design and operationalization of the desiderata. However, these are one-time costs and they are generalizable to different settings. For example, once we operationalized the novelty metric, we applied it to all capabilities domains (e.g., science, math, history, and more). These design costs of the desiderata can be amortized when we apply them to many downstream domains.
> > >
> > > **“The introduction and abstract could benefit from revisions to be more specific and descriptive.”**
> > >
> > > Could you point out which part of introduction lacks specificity? We would really appreciate more concrete editing suggestions. Thanks you!

---

> > > > ### Author Response · Authors · 2024-11-22
> > > >
> > > > **“What language model was used to generate Table 1? Adding this information to the caption would improve clarity.”**
> > > >
> > > > The set of models we evaluated on all the datasets are listed in Section 5.1 and Appendix D. It’s a relatively long list of 17 models, but we added a reference to the appendix in the footnote in this section.
> > > >
> > > >
> > > > **I am unclear about the procedure for qualitative assessment in Section 6.3. What does "high quality" mean, and how many question-answer pairs were sampled? What do the "ranks" in this section refer to?**
> > > >
> > > > Thanks for the questions, we have revised the writing to make it clear.
> > > >
> > > > - For capability datasets, high-quality means that the question is salient, and the answer is factually correct; for safety datasets, high-quality means the question is unsafe, and the answer provided by the candidate LM is also unsafe.
> > > > - We sample 2 Q,A pairs from each dataset. In Table 10 in the appendix, we included more samples.
> > > > - Recall that the novelty metric is computed based on rank correlation between Vc and \hat{Vc}. We discuss the ranking of candidate models on three capability datasets and two safety datasets generated by AutoBencher. For example, Gemini Pro attains a general ranking of 6 on MMLU, but it drops to 16 on our dataset about Fordism, explaining why it is novel performance patterns not captured by existing datasets.
> > > >
> > > > **How are the salience and harmfulness binary variables obtained in practice?**
> > > >
> > > > This can be found in line 336 of the paper. For knowledge-intensive datasets, we use the number of page views of the corresponding Wikipedia article. If the number of views exceeds 0.5M, the description is salient. For math and safety, we evaluate the categories by including humans in the loop to judge and prune the non-salient or non-harmful categories. Specifically, we ask humans to classify each description into two categories {salient/non-salient}. Empirically, we observe that the rejection rate for math and safety descriptions is close to 0, because prompted GPT-4 yields highly salient math/safety dataset descriptions.
> > > >
> > > > **Questions about separability and Novelty metrics.**
> > > >
> > > > We apologize for the confusion. Your interpretation is correct, we made a typo in the dimension of V_prev, and it should be M x N, and the \theta parameter has dimension N (the number of datasets).
> > > >
> > > > For the question about the separability metric. Yes, we are taking the mean over Vc.  Specifically, SEP(Dc,M) = mean(|Vc − mean(Vc)|). We agree that this metric does not provide the worse-case guarantee as the reviewer mentioned. To address this concern, we also include the metric for worst-case separability SEP_worsecase = min_{i,j} Vc[i] - Vc[j], which is the minimum pairwise distance in the accuracy vector, and this worse-case metric reveals the same high-level trend.
> > > >
> > > > |            | Separability (worst-case) | Separability (average-case) |
> > > > |-------------------|---------------------------|-----------------------------|
> > > > | **MMLU (HumanBench)** | 0.0                       | 0.011                       |
> > > > | **AutoBencher**       | 0.011                     | 0.026                       |

---

> > > > > ### Comment · Reviewer_bP3L · 2024-11-25
> > > > > **Re-comment**
> > > > >
> > > > > **How are the salience and harmfulness binary variables obtained in practice?**
> > > > >
> > > > > Let me me more clear here. If I were to use this method to generate datasets for, lets say, math, do I have to manually go through the descriptions?

---

> > > > ### Comment · Reviewer_bP3L · 2024-11-25
> > > > **Re-comment**
> > > >
> > > > **Cost analysis**
> > > >
> > > > Thanks for the clarification on computational costs. How do the computational costs scale with the various parameters of the method?
> > > >
> > > > I do believe cost efficiency is a major advantage of such methods. Emphasizing the cost efficiency of the method (compared to other ways of creating such datasets) can help highlight the strengths of this method.
> > > >
> > > > **Harmfulness**
> > > > Can you elaborate more on what you mean by this: "We found that 98.4% of the questions are labeled as harmful, and language models should abstain", and how the process can ensure that the generated questions are also not harmful?
> > > >
> > > > **The introduction and abstract could benefit from revisions to be more specific and descriptive.**
> > > >
> > > > Sincere apologies, let me be more specific:
> > > > - The abstract can specifically mention how you optimize by finding good dataset descriptions
> > > > - The abstract and intro claim that the generated datasets are "27% more novel and 22% more difficult" -- it is difficult to understand what this means without context. The metric for novelty, difficulty, etc. should at least be mentioned
> > > > - "In AutoBencher, we first declare a few desiderata for the dataset, then we build quantitative surrogate metrics
> > > > for them, and search for a particular dataset that optimizes an explicit objective of our desiderata. The
> > > > objective allows us to precisely measure the progress of our constructed datasets" -- I think it needs to be clearer in the introduction whether autobencher is trying to find one particular optimal dataset or a set of datasets
> > > > - It also needs to be clarified how the datapoints in the dataset consists of questions and answers grouped by topic

---

> ### Comment · Reviewer_bP3L · 2024-11-25
> **Re-comment**
>
> **Variance analysis**
>
> Thanks for the clarification -- I did not know that the temperature was set to 0 for the experiments. I don't remember if this was mentioned in the main paper or not, but if not it should be clarified.
>
> Also thank you for running additional experiments. I am not too concerned about the randomness with the proposing as I am with the scoring of generations.
>
> With regards generation, what is the variance of the (1) separability, (2) difficulty, and (3) novelty of different generational runs? How many (question, answer) pairs do you need to generate to get an accurate estimate of the criteria? Why was $|\mathcal{D}_c| = 50$ chosen as the sample size for the experiments?

---

> ### Author Response · Authors · 2024-11-25
> **Added Variance Analysis and Computational Cost Analysis**
>
> ## **Variance Analysis**
>
> Thanks for your suggestion! We added a new analysis and have edited the paper to include this.
>
> In the rebuttal, we performed three different generation runs conditioned on the same dataset description with three different random seeds at temperature 1. Now, we bootstrap from them to create different datasets of size N. We plot the standard deviation for novelty, separability and difficulty as a function of N.
>
> See the plot in the paper (appendix G, Figure 3), and also the anonymous link below. https://openreview.net/notes/edits/attachment?id=K4VGVZoJcK&name=supplementary_material
> Empirically, we observe that the curve looks proportional to $1 / \sqrt{n}$.
>
> Our final generated dataset contains at least 300 examples, and the standard deviation is (0.035, 0.016, 0.019) for novelty, separability and difficulty respectively.
>
> According to this plot, the standard deviation at 50 samples is roughly (0.095, 0.022, 0.039) for novelty, separability and difficulty respectively. This standard deviation defines an interval that excludes the HumanBench’s metrics in novelty, separability and difficulty. Specifically, both novelty and difficulty metrics are less than $\mu - 2 \sigma$. Therefore, 50 samples is roughly the lowest number of samples that we can get meaningful results compared with the human baseline. Once we selected the best dataset description, we run generation again to gather 300-500 examples, which brings our standard deviation down to  (0.035, 0.016, 0.019), yielding a robust estimate for the three metrics.
>
> ## **Computational Cost Analysis**
>
> Thanks a lot for the suggestion, and we appreciate that you like the cost-effectiveness of AutoBencher!
>
> Specifically, the hyperparameter that will impact computational cost is  A = the number of proposed datasets, B =  the number of examples per dataset, and C = the number of candidate LMs we evaluate.
>
> In the question generation phase, only the first two matters. The computational cost is roughly proportional to the total number of questions generated (A * B). Note that the context lengths of adaptive search will include historical metrics, which will be larger for each iteration. However, this delta of token cost (around 400 for each new iteration) is negligible compared with the cost of generating the full dataset.
>
> Based on our empirical estimation, **AutoBencher takes around 380 * A * B tokens (which costs 0.0075 * A * B dollars) to generate A * B questions**.
>
>
> In the model evaluation phase, the computational cost is proportional to A * B * C, because we need to evaluate every question on every model. In particular, since different candidate LMs have different parameter counts, the cost will also depend on the model size. In AutoBencher, we run inference of 7B models in an A100, and use API calls for larger models. For simplicity, I will use the price of GPT-4-turbo as an upper bound for API cost estimation, because it’s more expensive than or on par with most other models.
>
> The computational cost of evaluating a 7B model on 500 questions is 0.13 GPU hours. So, **the total GPU hour would scale based on (0.13 * A * B * C / 500)**.
>
> For inferring larger models (e.g., Claude, Gemini, GPT), the token cost also scales linearly with A*B*C. The average length of question+response is around 80 tokens for knowledge intensive questions. The API token count is **A * B*C * 80, and the cost is around A * B * C * 0.0016 dollars**.
>
> For judging the correctness of model responses (LLM-as-judge), it also scales linearly with A * B * C. Each LLM-as-judge query+response length is roughly 120 tokens. **The API token count isA * B * C * 120, and the cost is around A * B * C * 0.0024 dollars**.
>
>
> In summary, Let C1 be the number of 7B models we evaluated, and C2 be the number of API models we evaluated, and C = C1 + C2.
>
> GPU-hours: 0.00026 * A * B * C1
>
> API-credits: A * B * C2 * 0.0016 dollars
>
> API-credits (LLM-as-judge) =  A * B * C * 0.0024 dollars
>
>
> Note that the wall clock time will not scale linearly with A * B * C, because we can significantly parallelize these computations. All the API queries are independent, therefore they can all be parallelized, so adding more models or more data will not slow down the API calls, and we are not even close to the rate limit.
>
> ## **Harmfulness**
>
> For clarification, in the safety setting, the goal is to jailbreak the model, which entails finding the harmful queries that the models fail to reject [1, 2]. Therefore, we are explicitly trying to ensure the generation of harmful questions, and trying to minimize the occurrence of non-harmful questions.
>
> [1] Red Teaming Language Models with Language Models  https://aclanthology.org/2022.emnlp-main.225.pdf
>
> [2] Universal and Transferable Adversarial Attacks on Aligned Language Models https://arxiv.org/pdf/2307.15043

---

> > ### Author Response · Authors · 2024-11-25
> >
> > **"If I were to use this method to generate datasets for, lets say, math, do I have to manually go through the descriptions?"**
> >
> > Yes, this is correct based on the exact algorithm of AutoBencher in Algorithm 1 and that’s what we did. Empirically, each AutoBencher run will produce 40 descriptions (since we perform N = 8 iterations of adaptive search, each proposing K = 5 descriptions). Compared with manually curating benchmarks, judging these descriptions is significantly less labor-intensive.
> >
> >
> > Encouragingly, we observe that the salience rejection rate for math and safety descriptions is close to 0, which means we can further reduce the human labor cost. A cheaper algorithm design is to judge salience/safety after we perform reranking (after line 10-11). That way, you only need to judge the top 3 descriptions, and select the one that’s highest scoring subject to being salient. Even for a domain whose salience rejection rate is not very low, you could still go down the ranked list, and select the first salient description.
> >
> > We put the salience filtering step in line 4 in Algorithm 1 to accommodate for the knowledge intensive settings, where salience judgment is fully automatic, and salience rate is lower, this saves us the cost of wastefully generating datasets that’s not salient. For domains where salience judgment requires human-labor, we could switch to the algorithm proposed above, and move the filtering step to after line 11.
> >
> >
> > **Writing suggestions on abstract and intro**
> >
> > Thanks for clarifying the writing suggestions, we will accommodate these in the next version of the paper.

---

> > > ### Author Response · Authors · 2024-11-26
> > >
> > > We believe that we have addressed all the concerns in your review with additional ablation experiments, analysis, and discussion! Please let us know if there are any remaining concerns or questions. We would really appreciate it if you could consider raising the score!

---

> ### Comment · Reviewer_bP3L · 2024-11-27
>
> Thank you for your detailed response. I am largely satisfied with the variance and cost analysis.
>
> Can the authors update the paper with a revised abstract and introduction?
>
> Also, I believe this question from the original review was never fully answered:
>
> Are there any ablation experiments on the following: **the criteria for evaluating dataset descriptions**

---

> > ### Comment · Reviewer_bP3L · 2024-11-27
> >
> > With regards to raising the score, I am still concerned about the novelty/contribution and the human effort required during the process. However, as I am not the best judge of contribution, I will leave that to the other reviewers. Please do your best to address their concerns.
> >
> > I will raise my score once the paper has been revised and all the other reviewers concerns are largely addressed, especially concerns regarding contribution and comparisons to other methods.

---

> > > ### Author Response · Authors · 2024-11-28
> > >
> > > Thanks for the response!
> > >
> > > **Regarding revision**, we have revised the paper intro and abstract to address your comments.
> > >
> > > **Regarding the ablation studies about "the criteria for evaluating dataset descriptions"**
> > >
> > > In AutoBencher, we evaluate dataset description based on "salience" which measures the importance of a topic/description. Some potential alternatives could be "conceptual/topical novelty" of the topic/description, "conciseness" of the description etc, we believe that these are narrower than salience to serve as the general-purpose benchmark desiderata. Also, they would employ the same filtering-based algorithm, which means no algorithmic ablations.
> > > Therefore, we don't think that adding this baseline will contribute to the story of the paper. However, we appreciated all the other baselines you proposed and we have added them.
> > >
> > > **Regarding other reviewers, we believe we have addressed all of the other reviewers' concerns in our author responses and revisions.**
> > >
> > > For your concern about the novelty of the paper, other reviewers commented that novelty is a strength of this paper.
> > >
> > > - "On **novelty**, the approach generates human interpretable topics where model ranks exhibit surprising results."
> > >
> > > - "The two-step division (first generate topics and then generate datasets per topic) is especially **novel** and effective."
> > >
> > > - "The proposed approach has two primary **novel** aspects: the use of a tool (such as a Wikipedia knowledge database or a Python library for mathematical calculations) to avoid the evaluating LM from generating answers that could be incorrect, and an algorithm to generate the benchmark in a guided rather than brute-force manner."
> > >
> > > Regarding your concern about contribution, other reviewers also gave positive comments for both high-level framing and low-level execution.
> > >
> > > - (high-level) "The problem is formalized and packaged in an elegant and extensible way in the AutoBencher framework, and two important instances of the framework are studied."
> > >
> > > - (low-level) "Well-executed research methodology with manual validation of the discovered benchmarks"
> > >
> > > - (low-level) "The paper provides an in-depth empirical evaluation, uncovering vulnerabilities and assessing model performance. It also includes results from human evaluations to further validate the quality and relevance of the generated benchmark."
> > >
> > > - (low-level) "The qualitative examples [are] aligned with the experience of this reviewer when trying to manually jail-break LLMs, [this] being auto-discovered is encouraging"
> > >
> > >
> > > **Regarding comparisons to other methods.**
> > >
> > > None of the reviewers asked for a comparison with prior methods. This paper is a novel framework by formulating benchmark construction as an optimization problem, so there are no prior methods that are comparable.

---

> > > > ### Comment · Reviewer_bP3L · 2024-11-29
> > > >
> > > > Great, since the other reviewers have not responded, I will assume that their concerns are addressed. Happy thanksgiving!

---

### Official Review · Reviewer_A8iq · 2024-11-10

**Soundness:** 3
**Presentation:** 3
**Contribution:** 3
**Rating:** 5
**Confidence:** 5

**Summary:**

This paper presents AutoBencher, a declarative framework for automatic benchmark construction, and use it to scalably discover novel insights and vulnerabilities of existing language models. Specifically, it is instantiated in two scenarios: 1) capability evaluation and 2) safety evaluation.  In each scenario, a set of desirable characteristics is first formally defined. For capability evaluation, these are salience, difficulty, separability, and novelty. For safety evaluation, they are harmfulness and attack success rate.  Then, a language model is used to automatically construct descriptions of topics along with datasets in those topics, where a dataset is a set of (question, answer) pairs. Empirical results using GPT-4 as the evaluator show that the created datasets are on average 27% more novel and 22% more difficult than existing benchmarks. AutoBencher also helps identify specific gaps not captured by existing benchmarks: e.g., Gemini-Pro has knowledge gaps on Permian Extinction and Fordism while GPT-4o fails to decline harmful requests about cryptocurrency scams.

**Strengths:**

1. The problem of automatic benchmark generation in a guided manner is an important one. While LMs have been used as judges to automatically evaluate other LM's answers, this work proposes using LMs to also generate questions.

2. The problem is formalized and packaged in an elegant and extensible way in the AutoBencher framework, and two important instances of the framework are studied.  The two-step division (first generate topics and then generate datasets per topic) is especially novel and effective.

3. The proposed approach has two primary novel aspects: the use of a tool (such as a Wikipedia knowledge database or a Python library for mathematical calculations) to avoid the evaluating LM from generating answers that could be incorrect, and an algorithm to generate the benchmark in a guided rather than brute-force manner.

4. The empirical evaluation shows the promise of the proposed approach in generating datasets that are more novel and difficult than even hand-crafted ones like MMLU.

**Weaknesses:**

1. AutoBencher is currently stand-alone. The paper would be stronger if it integrated AutoBencher into existing popular evaluation frameworks like Stanford's HELM or HuggingFace's Open LLM. Adoption of AutoBencher by one of these frameworks would make a more convincing case for its usefulness and viability.

2. There seems to be a mismatch between the capabilities of the evaluating LM and the evaluated LMs: the former has access to tools whereas the latter does not. The paper does not make a convincing case why future LMs should be expected to have such capabilities without using tools themselves.

3. The evaluation is rather limited. I would expect a benchmark/evaluation focussed paper to be more comprehensive and derive more insights than those currently presented. For instance, AutoBencher currently only generates one-turn (question, answer) pairs; it would be interesting to see it extend to multi-turn data, chain-of-thought data, etc. Another direction to extend it could be in the domain of multi-modality.

4. I found the safety evaluation less convincing than the capability evaluation. While the paper does use recent baselines such as XSTest and HarmBench, it would be more convincing if the paper would report on how AutoBencher could be integrated into a mainstream framework for safety evaluation and discuss challenges that were overcome in such an integration (this is related to item 1 above but is more specific to safety evaluation).

**Questions:**

Please see Weakesses.

---

> ### Author Response · Authors · 2024-11-21
>
> We thank reviewer A8iq for their time and review. Reviewer A8iq primarily asked for incorporating AutoBencher into popular evaluation frameworks, and for a discussion on the role of privileged information. We addressed these comments below. Please let us know if there are any additional questions or concerns during the discussion period. Thank you!
>
>
> **“The paper would be stronger if it integrated AutoBencher into existing popular evaluation frameworks like Stanford's HELM or HuggingFace's Open LLM”**'
>
> We thank the reviewer for this suggestion to make AutoBencher more impactful.
> We have submitted pull requests to add AutoBencher-Capabilities and AutoBencher-Safety to the HELM repository, and they were accepted and incorporated into HELM.
>
>
>
> **“how AutoBencher could be integrated into a mainstream framework for safety evaluation?”**
>
>
> AutoBencher is adaptive, meaning that the datasets are generated conditioned on an LM under evaluation to reveal its safety vulnerabilities. Most safety evaluation framework is about red teaming, which is aligned with AutoBencher-safety’s goal and adaptiveness, so they can be easily incorporated. There are also some evaluation frameworks such as HELM that operate with static datasets. In order to produce a static dataset that is compatible with HELM, we combine the AutoBencher datasets for a set of candidate LMs to produce a static dataset that reveals model safety failures across the set of LMs we care about.
>
>
> **“The paper does not make a convincing case why future LMs should be expected to have such capabilities without using tools themselves.”**
>
> The key of automatic dataset construction is asymmetry, which doesn’t have to be in the form of tool use. For example, one form of asymmetry is providing **more test-time compute** to the evaluator LM. As shown by o1’s test time scaling result, more test-time compute can lead to better performance, leading to a stronger evaluator. Asymmetry could also rely on **the task structure where forward is easier than backward**. For example, randomly browsing the web to observe information is easier than actively seeking information [1]. We can leverage this gap to make the evaluator LMs generate questions that are hard to answer by the candidate LMs.
>
> Additionally, our constructed datasets are a useful intrinsic test of LMs capabilities. Even if we expect realistic downstream LMs applications to use RAG, knowledge-intensive benchmarks such as MMLU and TriviaQA still remain highly useful as capabilities tests of language models.
>
> We have added this discussion in the revised paper (Section E).
>
>
>
> **“I would expect a benchmark/evaluation focussed paper to be more comprehensive and derive more insights than those currently presented. For instance, AutoBencher currently only generates one-turn (question, answer) pairs; it would be interesting to see it extend to multi-turn data, chain-of-thought, and multi-modality.”**
>
> The reviewer made good suggestions about extending AutoBencher to multi-turn QA and multi-modality. We believe these are interesting potential extensions in future work. In this paper, we focused on single-turn QA in order to provide more extensive experiments on capabilities and safety. In this simple setting, we have already discovered meaningful insights, including the knowledge gap of Gemini and the safety vulnerabilities of GPT-4. Some of these automatically discovered insights echo human findings, which is very meaningful.
>
> [1] https://arxiv.org/abs/2403.08140

---

> > ### Author Response · Authors · 2024-11-26
> >
> > We believe that we have addressed all the concerns in your review by adding AutoBencher to HELM and adding additional discussions! Please let us know if there are any remaining concerns or questions.

---

### Author Response · Authors · 2024-11-22
**Summary**

We thank all of the reviewers for their helpful comments and suggestions. We’ve responded to each review separately and believe that we have addressed all reviewer concerns and questions. Please let us now if there are any other questions or concerns during the discussion period. Thank you!

We have strengthened the submission with additional experiments, discussions, and clarification based on the reviewer comments (changes in blue). We summarize the main changes below:

- We have included new ablation studies investigating the effect of privileged information in Appendix H.
- We have included new experiments investigating the variance and the effect of temperature in Appendix G.
- We have included new experiments investigating the impact of different evaluator LMs in Appendix I.
- We have included discussion about privileged information and computational costs in Section E and F.
- We have revised Section 3.1 to improve notations and content in 6.3 to clarify the qualitative studies.

---

### Meta-Review · Area_Chair_F9Tc · 2024-12-19

**Metareview:**

The paper "AutoBencher: Towards Declarative Benchmark Construction" introduces a novel framework, AutoBencher, for automatic benchmark construction using a declarative approach. AutoBencher leverages large language models (LLMs) to iteratively generate datasets optimized for specific desiderata, such as difficulty, novelty, and safety, revealing weaknesses and vulnerabilities in existing LLMs. The framework is applied to two primary scenarios: capability evaluation (e.g., math, multilinguality, knowledge) and safety evaluation (e.g., harmful prompts). Empirical results demonstrate that AutoBencher generates datasets that are more novel and challenging compared to existing benchmarks, identifying specific gaps not captured by current evaluations, such as knowledge deficiencies in models like Gemini-Pro and safety vulnerabilities in GPT-4o.

The primary contributions of the paper are:
- AutoBencher Framework : A declarative approach to benchmark construction that optimizes for specific desiderata (e.g., difficulty, novelty, safety) using LLMs, providing a scalable method to discover model weaknesses and vulnerabilities.
- Empirical Validation : Demonstrates that AutoBencher-generated datasets are 27% more novel and 22% more difficult than existing benchmarks, identifying specific knowledge gaps and safety vulnerabilities in state-of-the-art models.
- Asymmetric Evaluation : Utilizes privileged information (e.g., tools, Wikipedia data) to create challenging benchmarks, highlighting the importance of asymmetry in evaluation.
- Safety Evaluation : Identifies harmful prompts that existing models fail to decline, contributing to safety evaluation in LLMs.
Open-Source Integration : Submission of AutoBencher capabilities and safety datasets to HELM, enhancing its practical applicability.


#### Weaknesses
1. **Lack of Integration (A8iq)**: Initial absence of integration with popular frameworks like HELM, later addressed by authors.
2. **Capability Mismatch (A8iq)**: Concerns about the evaluator LMs' use of tools not available to evaluated models, somewhat mitigated by authors' clarifications.
3. **Limited Scope (A8iq)**: Focuses on single-turn QA pairs, with multi-turn and multi-modal datasets suggested as future work.
4. **Safety Evaluation (A8iq)**: Seen as less convincing, with integration into safety frameworks proposed but not empirically validated.
5. **Methodological Novelty (bP3L, oakz)**: Perceived as combining existing techniques without significant innovation, though authors highlight optimization-based construction and asymmetry.
6. **Practicality and Costs (bP3L)**: Concerns about computational intensity and manual effort, addressed with cost analyses.
7. **Variance (bP3L)**: Initial lack of variance analysis, resolved in rebuttal.
8. **Human Effort (bP3L, oakz)**: Manual salience judgments raise agreement issues, with solutions proposed but not implemented.
9. **Evaluator Bias (vFMj)**: Potential bias from using GPT-4 as evaluator, addressed with ablations.
10. **Qualitative Details (vFMj)**: Lack of detailed qualitative analysis, resolved in rebuttal.


The authors adequately address most of the questions during rebuttal, which is good.

**Additional Comments On Reviewer Discussion:**

1. **Integration with Evaluation Frameworks (A8iq):**
   - **Concern**: Lack of integration with existing frameworks like HELM or HuggingFace's Open LLM.
   - **Response**: The authors submitted pull requests to integrate AutoBencher into HELM, which were accepted, addressing the integration concern.
   - **Impact**: This response resolved the concern, demonstrating the framework's practicality and potential for adoption.

2. **Privileged Information and Model Capabilities (A8iq):**
   - **Concern**: Mismatch between evaluator and evaluated model capabilities due to tool usage.
   - **Response**: The authors clarified that asymmetry in capabilities can be achieved through additional compute or task structure, not just tool use, and emphasized the intrinsic value of benchmarks like MMLU for capability testing.
   - **Impact**: The clarification addressed the concern, though some reviewers still sought empirical validation of these claims.

3. **Limited Evaluation Scope (A8iq):**
   - **Concern**: Focus on single-turn QA pairs without exploration of multi-turn, chain-of-thought, or multi-modal datasets.
   - **Response**: The authors justified focusing on single-turn QA for extensive evaluation and acknowledged multi-turn and multi-modal extensions as future work.
   - **Impact**: The response was accepted as reasonable, though reviewers suggested these extensions could enhance the paper's scope.

4. **Safety Evaluation Concerns (A8iq):**
   - **Concern**: Lack of integration with mainstream safety evaluation frameworks and convincing safety results.
   - **Response**: The authors explained how AutoBencher's adaptive nature aligns with red teaming frameworks and proposed creating static datasets by combining AutoBencher outputs for multiple models.
   - **Impact**: The explanation provided a pathway for integration, though empirical validation in mainstream frameworks remains a potential area for improvement.

5. **Methodological Novelty (bP3L, oakz):**
   - **Concern**: Limited novelty beyond retrieval augmentation and optimization.
   - **Response**: The authors emphasized the novelty of optimization-based benchmark construction, asymmetric evaluation, and empirical findings about benchmark orthogonality.
   - **Impact**: The response highlighted the method's novelty, though some reviewers remained skeptical about its significance relative to existing techniques.

6. **Practicality and Computational Costs (bP3L):**
   - **Concern**: Potential computational intensity and manual effort in criteria design.
   - **Response**: The authors provided detailed cost analyses, emphasizing AutoBencher's scalability and efficiency compared to manual curation. They also discussed the one-time cost of criteria design.
   - **Impact**: The detailed analysis addressed concerns, though some reviewers still questioned the practicality of manual salience judgments.

7. **Variance and Robustness (bP3L):**
   - **Concern**: High variance in LLM outputs and lack of variance analysis in the original submission.
   - **Response**: The authors conducted additional experiments with different temperatures and seeds, showing high Pearson correlation across runs and robust metrics.
   - **Impact**: The variance analysis resolved the concern, demonstrating the robustness of AutoBencher's outputs.

8. **Human Effort in Salience Judgments (bP3L, oakz):**
   - **Concern**: Manual classification of dataset descriptions raises inter-annotator agreement issues.
   - **Response**: The authors proposed using majority voting or expert moderation for math salience judgments and noted the low rejection rate for prompted categories.
   - **Impact**: The proposed solutions addressed the concern, though reviewers suggested implementing these in practice.

9. **Bias in Evaluator Models (vFMj):**
   - **Concern**: Potential bias from using GPT-4 as the evaluator, favoring models from the same family.
   - **Response**: The authors conducted ablations with different evaluators (Claude-3.5, LLaMA 3.1-405B), showing consistent results and no significant bias.
   - **Impact**: The ablations resolved the bias concern, enhancing the paper's credibility.

10. **Qualitative Analysis Details (vFMj):**
    - **Concern**: Lack of detailed qualitative analysis and examples.
    - **Response**: The authors provided additional dataset examples and discussed model rankings and safety vulnerabilities in the rebuttal.
    - **Impact**: The additional details addressed the concern, though reviewers suggested including these in the main paper.


The authors' responsiveness to reviewer feedback, through additional experiments, clarifications, and integration efforts, strengthens the submission.

---

### Decision · Program_Chairs · 2025-01-22

Accept (Poster)